# Differential Accumulation of Misfolded Prion Strains in Natural Hosts of Prion Diseases

**DOI:** 10.3390/v13122453

**Published:** 2021-12-07

**Authors:** Zoe J. Lambert, Justin J. Greenlee, Eric D. Cassmann, M. Heather West Greenlee

**Affiliations:** 1Department of Biomedical Sciences, Iowa State University College of Veterinary Medicine, 1800 Christensen, Ames, IA 50011, USA; zlambert@iastate.edu; 2Virus and Prion Research Unit, National Animal Disease Center, Agricultural Research Service, United States Department of Agriculture, 1920 Dayton Ave., Ames, IA 50010, USA; justin.greenlee@ars.usda.gov (J.J.G.); eric.cassmann@ars.usda.gov (E.D.C.); 3Oak Ridge Institute for Science and Education, 1299 Bethel Valley Rd., Oak Ridge, TN 37830, USA

**Keywords:** bovine spongiform encephalopathy, cerebellar cortex, chronic wasting disease, Creutzfeldt–Jakob disease, enteric nervous system, Gerstmann–Straussler–Scheinker disease, Kuru, prion neuroinvasion, retinal ganglion cells, scrapie

## Abstract

Prion diseases, also known as transmissible spongiform encephalopathies (TSEs), are a group of neurodegenerative protein misfolding diseases that invariably cause death. TSEs occur when the endogenous cellular prion protein (PrP^C^) misfolds to form the pathological prion protein (PrP^Sc^), which templates further conversion of PrP^C^ to PrP^Sc^, accumulates, and initiates a cascade of pathologic processes in cells and tissues. Different strains of prion disease within a species are thought to arise from the differential misfolding of the prion protein and have different clinical phenotypes. Different strains of prion disease may also result in differential accumulation of PrP^Sc^ in brain regions and tissues of natural hosts. Here, we review differential accumulation that occurs in the retinal ganglion cells, cerebellar cortex and white matter, and plexuses of the enteric nervous system in cattle with bovine spongiform encephalopathy, sheep and goats with scrapie, cervids with chronic wasting disease, and humans with prion diseases. By characterizing TSEs in their natural host, we can better understand the pathogenesis of different prion strains. This information is valuable in the pursuit of evaluating and discovering potential biomarkers and therapeutics for prion diseases.

## 1. Introduction

Transmissible spongiform encephalopathies (TSEs) are a group of fatal, progressive neurodegenerative diseases that result from the misfolding and accumulation of endogenous prion protein (PrP^Sc^) [1]. Transmissible and spontaneous prion diseases have been described in a wide variety of species including humans [2], sheep [3,4,5], goats [6], mink [7], cattle [8], white-tailed deer [9], mule deer [9], elk [9,10] camels [11], moose [12], reindeer [13], red deer [14], sika deer [15], cats [16], and various zoo species [17,18].

A given prion disease within a species can have a range of phenotypes. Different phenotypes include, but are not limited to, differences in transmissibility within a species [19]; transmissibility between species [19]; disease time course [20]; most prominent clinical signs [21]; tissue, brain region [22] and cellular localization [23] of accumulated PrP^Sc^.

Often, disease phenotypes within a given species are denoted as different ‘strains’ [24]. Strains in a natural host are commonly defined by the molecular weight profile of the PrP^Sc^ [25], genotypes of susceptible hosts [23], age of disease onset [26,27], and distribution and intensity of misfolded PrP^Sc^ [20,28]. Because the molecular profile on a Western blot may be different depending on host species or strain of infection, to follow is a brief discussion of the significance. Following proteinase-K digestion of a sample homogenate containing misfolded prion protein, a Western blot results in three bands that are the di-, mono-, and unglycosylated isoforms (highest kDa to lowest kDa, respectively) of the prion polypeptide. Increased glycosylation adds weight, which slows migration of the prion protein. Thus, the diglycosylated band has the highest kDa, while the unglycosylated has the lowest kDa. Different strains of prion disease in a given species may have different molecular weight profiles, particularly of the unglycosylated band, due to the differential cleavage by proteinase-K of the misfolded protein’s n-terminal (reviewed in [29]).

Strains also may be grouped into what are referred to as ‘classical’ and ‘atypical’ based on proteinase K-resistant fragments, neuronal tropism, deposition patterns, and pathological lesion profile [30]. Classical cases of prion diseases tend to occur in groups of younger animals compared to atypical cases that tend to occur in individual older animals. Additionally, animals with classical cases of scrapie shed prions into the environment and thereby have a propensity for vertical and horizontal transmission under field conditions. Atypical cases of scrapie shed little to no prions into the environment, providing further evidence to support the spontaneous origin of atypical prion diseases. This, however, does not hold true in cases of cattle in which prions are not shed into the environment regardless of strain.

Rodent models have proven to be invaluable in isolating [31], identifying, and characterizing [32] strains of TSEs isolated from natural hosts [33]. However, to fully understand the pathogenesis of different TSE strains, they must also be characterized in their natural host. In many instances the character of PrP^Sc^ immunoreactivity as well as cellular and subcellular localization of accumulated PrP^Sc^ also differs between prion strains [34,35,36]. While Western blots provide information regarding protein size and strain following proteinase-K digestion, this review focuses on the differential localization and patterns of PrP^Sc^ that are observed using immunohistochemistry. In this review, we focus on PrP^Sc^ accumulation in three locations in the nervous system (retinal ganglion cells, cerebellar cortex, and enteric nervous system; see Figure 1). These three nervous system sites were selected based on the presence of robust literature that reports strain-dependent differential accumulation of PrP^Sc^ within these structures. We discuss their utility in differentiating strains and better understanding the pathogenesis of bovine spongiform encephalopathy in cattle, scrapie in sheep and goats, and chronic wasting disease in cervids.

## 2. Transmissible Spongiform Encephalopathies

### 2.1. Bovine Spongiform Encephalopathy

Bovine spongiform encephalopathy (BSE) was first described in cattle in 1986 [37]. BSE is grouped into classical and atypical strains [38]. After Western blotting, the molecular weight profile of the unglycosylated band in atypical strains either lower (L-BSE) or higher (H-BSE) than the classical strain (C-BSE). C-BSE is transmissible to cattle [8], sheep [39], goats [39], and humans [40,41] among other animals [18] via consumption of infectious material and is the agent responsible for the mid-1980s to mid-1990s epizootic disease in the United Kingdom where over 178,000 cattle were diagnosed positive for C-BSE [42,43] and millions of cattle were depopulated. As of 2021, 232 people have died from vCJD worldwide [44]. While C-BSE is infectious following ingestion, all known atypical BSE strains are either spontaneous [38] or are inherited via polymorphism in the PRNP gene [45]. Atypical strains of BSE and scrapie have been identified in older animals [46] and do not appear to be easily transmitted via the oral route [47]. While atypical scrapie and H-BSE do not seem to be infectious to humans [48,49], evidence suggests that L-BSE has the potential to be transmissible to humans [49]. Relative to other species with prion diseases, cattle with BSE generally have little to no lymphoid distribution of PrP^Sc^, except for tonsils and transiently in Peyer’s patches [50,51]. Though atypical strains appear in older animals, experimental intracranial transmission of atypical strains of BSE results in a much shorter incubation time compared to C-BSE [52,53]. Unlike other species, polymorphisms in the PRNP gene of cattle are extremely rare, thus disease susceptibility and incubation time is not affected by an animal’s genotype [53].

### 2.2. Scrapie

Scrapie is the prion disease of sheep and goats. The first records of scrapie date back to 1732 [54]. Following the first description of atypical (Nor-98) scrapie in 2003, discussion of scrapie strains is typically framed using classical and atypical scrapie in sheep [3]. There are differences in the molecular weight profile between classical and atypical strains of scrapie, as atypical scrapie has a smaller PrP^Sc^ fragment, is more PK sensitive, and has 5 less intense Western blot bands compared to the three strong bands in classical scrapie [3,4,26]. In sheep, classical scrapie is spread primarily via horizontal transmission shortly after animals are born, though the average age of affected animals is 2–5 years of age [55], and there is widespread distribution of PrP^Sc^ in lymphoid tissues [56,57]. Atypical scrapie in sheep appears to be sporadic, in that new cases appear in isolation, as opposed to a cluster of infected animals [3], and there is little to no accumulation of PrP^Sc^ in the lymphoid system [4,58]. There are a number of polymorphisms in the PRNP gene of sheep that affect susceptibility and incubation time of different strains of scrapie (reviewed in [26,59]). Experimental and epidemiological evidence suggests that there is an extremely low likelihood that scrapie could be transmitted to humans [29,60]. Still, studies show that this low likelihood leaves room for the possibility of scrapie transmission to humans [61,62].

### 2.3. Chronic Wasting Disease

Chronic wasting disease (CWD) is the prion disease of deer and elk [63]. CWD was first identified in 1967 and was formally described in 1980 in captive mule deer and black-tailed deer [64]. Several CWD strains have been described [31,65,66,67,68]. In cervids with CWD, there is widespread accumulation of PrP^Sc^ in lymphoid tissues [69]. There are several described polymorphisms in the PRNP gene that may influence disease susceptibility and incubation time [70]. CWD is extremely contagious between cervids, and while there is some evidence it is transmissible to other species [71,72,73,74], there is no evidence that it has been transmitted to humans [9,75]. 

### 2.4. Human Prion Diseases

The most common prion disease in humans is Creutzfeldt–Jakob disease (CJD). CJD consists of multiple strains that include iatrogenic, variant, familial, and sporadic CJD (iCJD, vCJD, fCJD, and sCJD, respectively). The basis of these strains is their origin: acquired, inherited, or spontaneous [76]. For example, iCJD and vCJD are acquired. The cause of iatrogenic transmission of CJD is a surgical or medical procedure involving materials (pituitary hormones, dura mater graft, blood transfusion, etc.) contaminated with PrP^Sc^ [76], while the variant strain is due to the ingestion of the misfolded prion protein from contaminated beef products [76]. Familial CJD is heritable and is the result of mutations at codons in the PRNP gene [76]. Lastly, sporadic CJD is spontaneous [76]. 

In 1974 the first case of iCJD was reported. This case was the first of many in which patients underwent a medical procedure with contaminated tissues or instruments and an incubation period of years to decades would follow. Due to subsequent public health and decontamination measures, iCJD does not pose the threat that it once did [77]. The first case of the vCJD epidemic in the United Kingdom was diagnosed in 1996, and evidence supports its cause being the consumption of cattle that were infected with BSE [40,41]. Susceptibility to vCJD is influenced by polymorphisms at codon 129 of the PRNP gene. For example, almost all vCJD cases are homozygous for methionine at codon 129 (MM129) [78,79,80]. Depending on the source, only one or two heterozygous individuals (MV129) have been reported to have vCJD [78,81,82], and no one homozygous for valine has reported to have acquired vCJD [78,82]. 

Prion diseases in humans can be caused by inherited mutations in the PRNP gene, the most common mutation causing familial CJD is E200K. The number (200) refers to the codon of the PRNP at which a mutation occurs. The letters (E and K) refer to the one-letter code for amino acids (glutamic acid and lysine). In the case of E200K, the resulting amino acid is E (glutamic acid), rather than K (lysine), at codon 200 of the PRNP gene. Occasionally, E200K-129M may be used. This indicates that the individual is homozygous for M (methionine) at codon 129 of the PRNP gene. Familial CJD is an inherited form of prion disease caused by mutations in the PRNP gene. As mentioned, the most common mutation is E200K [83], which is homologous [45,53] to the polymorphism observed in cattle with the hereditable E211K H-BSE [45]. At least 14 other autosomal dominant mutations causing fCJD have been reported [84]. 

Sporadic CJD has the greatest phenotypic variety and is genotype-dependent, as the majority of sCJD cases are homozygous for methionine (MM) at codon 129 of the PRNP gene [78]. The genotype of codon 129 of the PRNP gene and type of prion protein accumulation underlies the differentiation of subtypes of sCJD [85,86,87,88]. At codon 129 of the PRNP gene, people can be either heterozygous for methionine and valine (MV129) or homozygous for either methionine (MM129) or valine (VV129). Further, there are two types of prion accumulation that are designated Type 1 and 2. Type 1 prion accumulation has a PK cleavage site at residue 82 [86], a PK-resistant core of 21 kDa [85], and has a smaller aggregation size following filtration (mean pore size 72 nm) than Type 2 prion aggregation size [87]. Type 2 has a PK cleavage site at residue 97 [86], a PK-resistant core of 19 kDa [85], and has a larger aggregation size following filtration (mean pore size 72 nm) than Type 1 prion aggregation size [87]. In all, cases of sCJD can be broken down into the following pure subtypes based on genotype at codon 129 of the PRNP gene and type of prion accumulation: MM1, MV1, VV1, MM2, MV2, and VV2 [88,89,90]. Concurrent Type 1 and Type 2 prion accumulation occurs in over one-third of sCJD cases [88]. Prion strains can be differentiated by ratio of di-, mono-, and unglycosylated protein as well as relative molecular mass [91]. Interestingly, variants of sCJD can also present with visual disturbances, as people the Heidenhain variant of sCJD present with visual symptoms at disease onset along with early posterior cortical involvement [92].

Humans are susceptible to other prion diseases, such as Kuru, fatal familial insomnia (FFI), and Gerstmann–Straussler–Scheinker (GSS). Kuru, similar to vCJD, is an acquired prion disease. It is specific to the Fore people in New Guinea whose members participated in ritual cannibalism as a means of mourning and respect for their deceased kindred [93]. In the Fore language Kuru means trembling or shivering, which reflects the symptom onset of cerebellar ataxia and tremor [93]. Prevalence of the disease has dramatically decreased due to the termination of the practice [93]. FFI, similar to fCJD, is a hereditary prion disease due to a mutation at codon 178 of the PRNP gene that substitutes asparagine with aspartic acid (D178N) and is autosomal dominant [94]. Lastly, GSS is also a hereditary prion disease and can be caused by several mutations. These mutations include P102L (most common) [83], P105L, A117V, F198S, Q217R, Q212P, and D202N [95]. Mutations causing GSS are autosomal dominant. It has been reported that people with GSS may have symptomatic abnormal eye movements and optic atrophy, the latter of which is extremely rare in people with CJD [96,97,98,99].

## 3. Retinal Ganglion Cells

The retina displays functional and morphologic changes that are associated with protein misfolding diseases. Accordingly, the retina is of major interest to identify potential biomarkers for proteinopathies and neurodegenerative diseases, such as Parkinson’s disease, Alzheimer’s disease, Huntington’s disease, and multiple sclerosis [100,101,102,103,104,105,106,107,108]. As such, the retina in cases of prion disease is also being studied [100,109]. It has been demonstrated that PrP^Sc^ accumulates in the retina of cattle with BSE [47,52,53,100,110,111,112], sheep and goats with scrapie [58,109,111,113,114,115,116,117,118,119,120,121,122,123,124], cervids with CWD [121,125,126,127,128,129,130,131,132,133], and humans with prion diseases [134,135,136,137]. 

The eye is the most accessible part of the central nervous system. Thus, it can be imaged and functionally assessed using noninvasive techniques. Within the eye, the retina is responsible for transmitting visual stimuli to the brain for processing [138]. The retina is organized into functionally distinct cellular and synaptic layers of cells. PrP^Sc^ differentially accumulates the retinal layers of animals with TSEs depending on factors ranging from strain of TSE to host genotype [52,53,100,119,125,136,139,140]. Once photons of light are transduced by the photoreceptor cells in the outer nuclear layer, the signal is transmitted to the inner nuclear layer and then the ganglion cell layer via the inner and outer plexiform layers. It is the axons of the retinal ganglion cells in the ganglion cell layer that exit the globe as the optic nerve to connect with subsequent parts of the visual pathway. Animals with TSEs accumulate PrP^Sc^ in the retina and multiple parts of the visual pathway [109,125,141]. Within the cellular and synaptic layers of the retina, PrP^Sc^ accumulation is most common in the plexiform layers of the retina [115,120,121,126,139,142,143] while the retinal ganglion cells demonstrate a strain-dependent variation of PrP^Sc^ accumulation [52,53,58,100,125]. 

There are strain-dependent differences in retinal function, prion accumulation, glial response, autophagic response, neuroinflammation, and other morphologic changes in the retinas of animals with TSEs [52,100,110]. Electroretinograms and optical coherence tomography have been used to assess functional and morphologic differences between strains of prion disease in cattle [100], sheep [109], and goats [122]. While electroretinograms measure the retina’s response to visual stimuli, optical coherence tomography can be used to measure retinal thickness. The earliest functional difference associated with the accumulation of pathological prion protein is present in electroretinogram data 11 months prior to clinical signs of disease in cattle with BSE [100]. Strain-dependent functional differences include prolonged b-wave implicit time [53,100,110,144] beginning at 12 months post inoculation [100] and continuing until clinical endpoint in cattle with C-BSE and H-BSE [100]. Additional data from optical coherence tomography shows retinal thinning in cattle [53,100,110] that is first detectable at 12 months post inoculation in cattle inoculated with C-BSE and H-BSE [100] and continues until clinical endpoint [100,110]. Strains of BSE can also be differentiated by the types of immunostaining for PrP^Sc^. In sheep, functional differences are evident by electroretinogram in the reduction of a-wave and b-wave amplitudes [109]. While not statistically significant, the average retinal thickness measured using optical coherence tomography in goats with scrapie trended lower than their scrapie-free counterparts [122]. PrP^Sc^ accumulates in the retina of cervids with CWD; however, functional data are not available [121,125,127]. Electroretinogram data demonstrates that people with CJD have a significant reduction in b-wave amplitude [145,146]. Here we discuss the differential PrP^Sc^ accumulation in the retinal ganglion cells (summarized in Table 1, Table 2, Table 3 and Table 4).

### 3.1. Bovine Spongiform Encephalopathy

Generally, retinal tissues from cattle experimentally inoculated with C-BSE have less immunoreactivity for PrP^Sc^ as compared to H-BSE, E211K H-BSE, and L-BSE [52,53,100,147]. These strain-dependent differences in the extent and intensity of PrP^Sc^ immunoreactivity in the retinas of cattle are illustrated and summarized in Figure 2A–D and Table 1. The deposition pattern in the retinas of cattle inoculated with C-BSE has been described as ranging from punctate and granular [52] to multifocal coalescing granular and globular [100] staining that is localized to the plexiform layers. Retinal tissues from cattle with atypical BSEs have more PrP^Sc^ immunoreactivity compared to C-BSE [38,52]. PrP^Sc^ immunoreactivity is more extensive, intense, and uniform in the retinas of cattle with atypical BSEs compared to cattle with C-BSE, especially in the plexiform layers [52,100]. While cattle with C-BSE have little to no PrP^Sc^ immunoreactivity outside the plexiform layers, cattle with H-BSE have staining in all retinal layers. In addition to globular deposits of misfolded prion protein in the photoreceptor cells, it also is reported that there are multifocal globular deposits in the nuclear layers of the retinas of cattle with H-BSE [52,100]. Cattle with H-BSE have intense granular PrP^Sc^ accumulation in the plexiform layers [112]. Further differentiation between C-BSE and H-BSE can be conducted using the 12B2 antibody that is immunoreactive to tissues affected by H-BSE and not C-BSE [100]. Using the 12B2 antibody, tissues from cattle with C-BSE cannot be differentiated from control animal tissues while those with H-BSE displayed immunoreactivity throughout the retina [100]. The deposition pattern of PrP^Sc^ in retinal ganglion cells of cattle experimentally inoculated with H-BSE is similar to that of cattle inoculated with E211K H-BSE, both demonstrating prominent intracellular accumulation [53]. Deposition of PrP^Sc^ is intense and abundant in the plexiform layers in cattle inoculated with E211K H-BSE (Figure 2D) [53,110]. In one study using three isolates of L-BSE (French low-type, Canadian low-type, and Italian BASE), no differences in PrP^Sc^ deposition in the retina were reported between these isolates of L-BSE [52,148]. Similar to H-BSE, the deposition pattern in cattle experimentally inoculated with L-BSE is greater in amount and distribution compared to cattle inoculated with C-BSE [38,52]. Similar to H-BSE, the retinas affected by L-BSE have intense punctate deposition that is reported to be more intense and uniform in the plexiform layers compared to those of C-BSE [52,100]. Compared to cattle with H-BSE, the retinal layers of cattle with L-BSE have fewer deposits of PrP^Sc^; however, these deposits are still greater than cattle with C-BSE [52]. Cattle with L-BSE demonstrate intense punctate PrP^Sc^ accumulation in the plexiform layers of the retina [144]. 

The greatest difference in staining between strains of BSE can be observed in the retinal ganglion cells. Intraneuronal staining of the retinal ganglion cells in cases of cattle with C-BSE is less than atypical cases [52,53,100]. Cattle with H-BSE have intense granular PrP^Sc^ accumulation in the ganglion cell layer [112] along with hallmark intense globular [100] and robust intracellular [52] PrP^Sc^ deposits present in the retinal ganglion cells (Figure 2C) [52,100]. Deposition of PrP^Sc^ is a prominent feature in the cell bodies of the retinal ganglion cells in cattle inoculated with E211K H-BSE (Figure 2D) [53,110]. Importantly, the retinal ganglion cells of cattle with L-BSE have intense punctate [144] and robust intracellular [52] deposits of misfolded prion protein (Figure 2B). Overall, PrP^Sc^ accumulation is differential in the cell bodies of retinal ganglion cells of cattle with atypical BSEs that is increased compared to cattle with C-BSE.

### 3.2. Scrapie

There is differential accumulation of misfolded prion protein in the retinas of sheep with classical and atypical scrapie. These differences are best demonstrated in Figure 2E–H and Table 2. Of the two US isolates of classical scrapie in sheep, No. 13-7 and x124 [59,115,149], no differences were reported in the PrP^Sc^ deposition patterns between retinas [115]. In sheep oronasally inoculated with the No. 13-7 classical strain, sheep homozygous for lysine at codon 171 (KK171) of the PRNP gene were resistant, while sheep with glutamine at codon 171 (QK171 or QQ171) accumulated PrP^Sc^ in the retina [119]. The plexiform layers of the retinas in sheep with classical scrapie have confluent, punctate, globular, intense, and coarse particulate deposition patterns [109,113,114,115,116,118,124]. In sheep with classical scrapie, the photoreceptor cell layer has multifocal punctate [114,115,118] deposits with granules in the inner segments of the photoreceptor cell layer [109,113,116,150]. Interestingly, sheep of AA136 genotype and oronasally inoculated with x124 classical scrapie were resistant while those intracranially inoculated accumulated PrP^Sc^ in the retina [115]. The retinas of the No. 13-7-inoculated counterparts of this study all had immunoreactivity for PrP^Sc^ [115]. Evidence suggests that PrP^Sc^ accumulates differently in the retinas of sheep with atypical scrapie compared to those with classical scrapie [58] although the retina is not otherwise reported. In the retinas of sheep with atypical scrapie there is prominent immunoreactivity for PrP^Sc^ in the plexiform layers with minimal staining in the nuclear layers (Figure 2H) [58]. Results of immunohistochemical staining of the retina were not reported in other strains of scrapie such as CH1641 and SSBP1 [151]. In the retinas of goats with classical scrapie, the PrP^Sc^ deposition pattern reflects what is observed in sheep with classical scrapie [122,123]. PrP^Sc^ accumulation is intense and extensive the goats with classical scrapie [122,123]. Along these lines, there is strong [123] particulate [122] accumulation in the plexiform layers and weak accumulation in the nuclear layers [123].

Retinal tissues from sheep with classical scrapie have PrP^Sc^ immunoreactivity in the cell bodies of retinal ganglion cells while RGC immunostaining negligible in sheep with atypical scrapie. The ganglion cell layer in sheep with classical scrapie has multifocal punctate deposits of PrP^Sc^ [109,113,114,115,116,118] with granular [109,124] and coarse particulate [116,124] deposits occurring in the cell bodies of retinal ganglion cells [113] (Figure 2E–G). In the retinas of sheep with atypical scrapie, there is minimal staining in the ganglion cell layer (Figure 2H) [58]. In goats with classical scrapie, there is intense accumulation in the ganglion cell layer [122] and strong prominent cytoplasmic deposition in retinal ganglion cells [123]. Similarly, intense intraneuronal PrP^Sc^ accumulation is reported in the retinal ganglion cells of goats with classical scrapie (Figure 2G) [122,123]. In summary, increased PrP^Sc^ accumulation occurs in the retinal ganglion cells of sheep and goats with classical scrapie and not in those with atypical scrapie.

### 3.3. Chronic Wasting Disease

PrP^Sc^ accumulates in the retinas of cervids with CWD [121,125,127,128,129,130,131,132,133] and is different depending on the species affected by CWD and genotype within a species. For example, the degree of PrP^Sc^ accumulation in the retinas of Rocky Mountain elk [125,130,132] is genotype- and strain-dependent based on the polymorphism at codon 132 of the PRNP gene [125,130]. These differences are best demonstrated in Figure 2I,J and Table 3. The most common genotype at codon 132 in captive and free-ranging Rocky Mountain elk is homozygous methionine (MM132) [125]. When elk of the MM132 genotype have CWD, their retinas display intense staining in the plexiform layers and no intraneuronal PrP^Sc^ immunoreactivity in the retinal ganglion cells [125,130]. The retinas of Rocky Mountain elk heterozygous at codon 132 of the PRNP gene (ML132) demonstrate intracytoplasmic accumulation of PrP^Sc^ [125]. Rocky Mountain elk homozygous for leucine at the codon 132 of the PRNP gene (LL132) have heavy intraneuronal PrP^Sc^ accumulation in the cell bodies of retinal ganglion cells compared to MM132 (Figure 2I,J) [130]. Passage of the CWD agent through LL132 elk may result in a strain separate than what is isolated from ML132 and MM132 elk [31]. Overall, retinal tissues from Rocky Mountain elk with the LL132 strain have more extensive PrP^Sc^ immunoreactivity when compared to other genotypes and strains. 

Other cervids with CWD also accumulate PrP^Sc^ in the retina. Misfolded prion protein accumulates in the retinas of white-tailed deer [127,128,129,133], reindeer [131,132], and mule deer [121]. Within each species, no differential PrP^Sc^ accumulation has been reported between retinas in a genotype- or strain-dependent manner. Reindeer accumulate PrP^Sc^ in the retina [131,132] with intraneuronal immunoreactivity in the retinal ganglion cells [132]. Immunoreactivity for PrP^Sc^ in the retinas of reindeer can be punctate, particulate, and coalescing deposits in the plexiform layers with scattered intramicroglial deposits [132]. Mule deer accumulate PrP^Sc^ in the retina [121]. While prion accumulation is reported in the plexiform and ganglion cell layers of mule deer [121], there has been no further characterization of this staining. Polymorphisms exist at codons 95 and 96 of the PRNP gene in white-tailed deer, but there were no differences in PrP^Sc^ immunoreactivity based on genotype [127,128,129,133]. PrP^Sc^ staining patterns in the retinas of white-tailed deer are diffuse granular in the plexiform layers [127,129] and fine multifocal in the ganglion cell layer with little to no accumulation in the retinal ganglion cells as seen in Figure 2K [128]. In contrast, white-tailed deer inoculated with the sheep scrapie agent have robust accumulation of PrP^Sc^ in the cell bodies of retinal ganglion cells (Figure 2L) [152]. This is notable because it provides a way to differentiate CWD from scrapie in white-tailed deer.

### 3.4. Human Prion Diseases

PrP^Sc^ accumulation has been demonstrated in the retinas of people with CJD [134,135,136,137]. These differences are summarized in Table 4. In cases of the sporadic, variant, familial, and iatrogenic strains of CJD, PrP^Sc^ accumulation occurs primarily as strong uniform staining in the plexiform layers [134,135,136]. In people with the most common subtype of sCJD (MM1), the plexiform layers are reported to be immunoreactive in a focal granular pattern [134]. There is no intraneuronal staining of the retinal ganglion cells reported in people with sCJD and vCJD [135,136]. PrP^Sc^ accumulation is both genotype- and strain dependent because people with sCJD of the most common subtype (MM1) have less PrP^Sc^ accumulation in the retina compared to both people with sCJD of a less common subtype and vCJD [134]. Supporting strain-dependent differential PrP^Sc^ accumulation, people with vCJD have a higher concentration of PrP^Sc^ in the retina relative to brain than people with sCJD [137]. The accumulation of PrP^Sc^ in the retina has not been reported in studies of people with GSS [95,153,154] or Kuru [155,156,157,158].

## 4. Cerebellar Cortex and White Matter

Depending on the strain of TSE and host genotype, misfolded prion protein accumulates differentially in the cerebellar cortex of cattle with BSE [53,159,160], sheep and goats with scrapie [23,26,29,161], cervids with CWD [129,132,162], and humans with CJD, GSS, and Kuru [95,158,163]. These strain-dependent differences include the intensity of PrP^Sc^ accumulation relative to the brainstem at the level of the obex [23,26,29,160,161], immunolabelling pattern [129,159,161], cerebellar cortex layer with greatest PrP^Sc^ accumulation [26,129,159], and molecular profile [110]. Within the cerebellum, the cerebellar cortex consists of three cellular layers: the molecular layer, Purkinje cell layer, and granule cell layer. Immediately deep to the granule cell layer of the cerebellar cortex is the cerebellar white matter that allows axonal fibers to enter and exit the cerebellum for communication with the cerebrum and body. 

The role of the cerebellar cortex is to receive and integrate information from the brain and body in order to produce coordinated goal-directed movements as well as maintain posture and balance. This role of the cerebellar cortex includes correcting errors in voluntary movement via receiving signals from the cerebrum and feedback from the body, which it can then integrate into appropriate efferent signals to ultimately synapse with somatic muscle fibers. Lesions in the cerebellum manifest as errors in goal-directed movement and posture. For example, animals or people may present with uncoordinated flexor and extensor muscles, ataxia, or hypermetria. These manifestations of cerebellar dysfunction are evident clinical signs in animals and people with prion diseases [4,58,164,165,166]. However, clinical signs do not always reflect the level of PrP^Sc^ accumulation in the cerebellum. For example, goats with scrapie may accumulate a similar level of PrP^Sc^ in the cerebellum compared to sheep; however, they may not present with cerebellar signs [164]. Still, the molecular and granule cell layers of the cerebellar cortex display differential accumulation of misfolded prion protein. Here, when discussing the intensity of misfolded prion accumulation in the cerebellum, it is always relative to the brainstem at the level of the obex. We summarize the differential accumulation that occurs in the cerebellar cortex of animal and humans with prion disease (Table 5, Table 6, Table 7, Table 8 and Table 9).

### 4.1. Bovine Spongiform Encephalopathy

There are differences in the intensity and patterns of PrP^Sc^ accumulation in the cerebella of cattle with different strains of BSE [47,53,110,112,159,160,167,168,169]. A summary of these differences can be found in Table 5. In cattle with C-BSE, there is more PrP^Sc^ deposition in the molecular layer than in the granule cell layer [53]. In the molecular layer there is prominent stellate and linear staining patterns whereas the granule cell layer is made up of fine and coarse granular to aggregated staining patterns [53] [53,167,168,170]. In relation to the brainstem at the level of the obex, the cerebellar cortices of cattle with L-BSE have significantly stronger PrP^Sc^ signal intensities [47,160] than cattle with C-BSE and H-BSE [160]. Cattle with L-BSE have an even distribution of immunoreactivity in the molecular and granule cell layers [159]. The PrP^Sc^ accumulation in the cerebellum of cattle with L-BSE was diffuse and is a similar pattern to the accumulation observed in the cerebellar cortices of sheep with atypical scrapie [159]. Compared to cattle with L-BSE, the molecular and granule cell layers of cattle with H-BSE have substantially less PrP^Sc^ accumulation that is less uniform [159]. Cattle with H-BSE have a stellate immunolabelling of PrP^Sc^ with plaques in the cerebellar cortices [112] and pronounced widespread glial staining in the white matter of the cerebellum when compared to C-BSE and L-BSE counterparts [112,159]. In the E211K H-BSE, staining of the cerebellum is scant with small, multifocal clumps of PrP^Sc^ in both the molecular and granule cell layers [110]. Further, there is a fine granular, particulate, and stellate immunolabelling pattern in the molecular layer with coarse granular and particulate labelling of PrP^Sc^ in the granule cell layer [53]. Cattle with the E211K H-BSE generally do not display immunoreactivity in the cerebellar white matter, differing from cattle with H-BSE in which the most prominent PrP^Sc^ staining occurs in the cerebellar white matter [110,159]. In cattle with E211K H-BSE, immunolabelling against PrP^Sc^ in the granule cell layer is comparable to that of C-BSE. Cattle with E211K H-BSE have fine granular and stellate labelling in the molecular layer of the cerebellum while cattle with C-BSE have prominent stellate and linear labelling in the molecular layer [53]. Little information is specifically reported on the Purkinje cells of cattle with BSE; however, published images indicate that there is little to no staining of Purkinje cells in any strain of BSE [53,159]. Overall, the greatest difference in the cerebellar cortices of cattle with different strains of BSE is L-BSE in which there is a significantly greater concentration of PrP^Sc^ accumulation in relation to the brainstem at the level of the obex. These differences suggest that the cerebellum is more reliable by immunohistochemistry for differentiating between strains of BSE than the brainstem at the level of the obex in cattle with BSE [159].

### 4.2. Scrapie

The relative intensity and pattern of PrP^Sc^ accumulation varies in the cerebella of sheep and goats with different strains of scrapie [4,26,58,119,122,123,161,164,171,172]. A summary of these differences can be found in Table 6 and Table 7. Overall, PrP^Sc^ deposition in the cerebellum (relative to the brainstem at the level of the obex in the same animal) is less intense and widespread in cases of classical scrapie compared to cases of atypical scrapie [26]. In sheep with classical scrapie, the cerebellum has strong immunoreactivity for PrP^Sc^ that is stronger in the granule cell layer than the molecular layer [26]. The staining is multifocal in the molecular layer and white matter [26]. The Purkinje cells in sheep with classical scrapie have some intraneuronal PrP^Sc^ accumulation [119]. The staining in the cerebella is genotype-dependent at codon 171 of the PRNP gene in sheep with classical scrapie [119]. Sheep with lysine at codon 171 of the PRNP gene (QK171 or KK171) had similar staining in the cerebella and was different from sheep homozygous for glutamine (QQ171) [119]. The former commonly have accumulation of misfolded prion protein in the white matter while the latter sheep lack this PrP^Sc^ accumulation [119]. Overall, the patterns of accumulation in the granule cell and molecular layer in the cerebellar cortices in classical scrapie is not genotype-dependent at codon 171 of the PRNP gene: granular, intraneuronal, and intraglial in the granule cell layer as well as scant punctate, granular, and stellate patterns in the molecular layer [119]. 

In sheep with atypical scrapie, the cerebellum has intense immunoreactivity for PrP^Sc^ that is stronger in the molecular layer than the granule cell layer [26,58,173]. Sheep with atypical scrapie primarily have PrP^Sc^ accumulation in the cerebellar cortex [3,4,58,161,171,173] while sheep with classical scrapie do not [4,26]. In sheep with atypical scrapie the staining patterns in the cerebellum is granular and punctate [58,173]. Cases of atypical scrapie have varied staining in the cerebellar white matter [161]. Unlike cases of classical scrapie in sheep, the Purkinje cells of those with atypical scrapie are always negative [161]. Again, atypical cases of scrapie have higher relative amounts of misfolded prion accumulation in the cerebellum in relation to their brainstem compared to the same structures in sheep with classical scrapie [58]. Still, some diversity exists in the degree of PrP^Sc^ deposition, as it has been reported that some sheep with atypical scrapie have cerebella that are minimally affected [171]. When classical scrapie presents in goats, there is PrP^Sc^ accumulation in the cerebellum [122,123,164]. Subpial immunolabelling against PrP^Sc^ occurs in the cerebella of goats with classical scrapie [122]. Cytoplasmic staining is strong and widespread in the molecular layer, granule cell layer, Purkinje cells, and cerebellar white matter [123]. In sheep with scrapie, the cerebellum allows for differentiation between strains. While the cerebellar cortex displays PrP^Sc^ immunoreactivity in sheep with classical and atypical strains of scrapie, it is greater in sheep with atypical scrapie in relation to PrP^Sc^ accumulation in the brainstem at the level of the obex. Additionally, sheep with classical scrapie tend to accumulate more misfolded prion protein in the granule cell layer while sheep with atypical scrapie accumulate more PrP^Sc^ in the molecular layer.

### 4.3. Chronic Wasting Disease

The misfolded prion protein deposits differentially in both intensity and pattern in the cerebella of cervids with CWD, as species and genotype both affect PrP^Sc^ accumulation [129,132,162,174]. A summary of these differences can be found in Table 8. The accumulation of misfolded prion protein differs in the cerebella of white-tailed deer based on genotype. In experimentally inoculated white-tailed deer, there are two PRNP codons associated with differences in PrP^Sc^ accumulation. The white-tailed deer wild-type prion protein allele is Q95/G96; codon 95 can instead be histidine (H) and codon 96 can instead be serine (S). Differences in accumulation pattern and intensity of PrP^Sc^ will be discussed in the following order: wildtype, S96, H95, H95/S96. In wildtype white-tailed deer, there is abundant coalescing PrP^Sc^ and plaques stretching in the granule cell layer and Purkinje cell layer. This occurs as coarse granular and large plaques [129]. In deer of the S96 genotype, accumulation of PrP^Sc^ is confined to the granule cell layer and white matter. In comparison to wildtype white-tailed deer, deer of the S96 genotype display less intense accumulation that is granular and diffuse [129]. Even less PrP^Sc^ accumulation occurs in the cerebellar cortices of deer of the H95 genotype that are infected with CWD [129] with the predominance of staining in the granule cell layer and minimal labelling in the Purkinje cell and molecular layers. This accumulation of PrP^Sc^ is discontinuous and diffuse in the granule cell layer as fine punctate and coarse small granular deposits with some plaque-like deposits [129]. When these polymorphisms are simultaneous (H95/S96), accumulation of PrP^Sc^ manifests as fine punctate and coarse granular deposits that were evenly distributed in the granule cell layer [129]. The molecular layer of white-tailed deer with H95/S96 polymorphisms demonstrated more intense accumulation of PrP^Sc^ than the deer of the wildtype, S96, and H95 genotypes [129]. White-tailed deer of the H95/S96 genotype displayed stellate aggregates of misfolded prion protein in the molecular layer of the cerebellar cortex [129]. 

In reindeer, PrP^Sc^ accumulation ranges in intensity overall [132], but is consistently more intense in granule cell layers when compared to the molecular layer [13,132] with Purkinje cells devoid of accumulation [132]. Reindeer show punctate deposits of PrP^Sc^ throughout the cerebellar cortex in less intense cases while particulate and aggregated deposits occur in more intense cases [132]. The cerebellar cortex of Rocky Mountain elk affected by CWD has glial-associated PrP^Sc^ accumulation that was granular to punctate and was most prominent in the white matter [130]. Mule deer with CWD have accumulation of PrP^Sc^ plaques in the cerebellar cortex that are present in the granule cell layer and molecular layer [174]. While the cerebellar cortex offers discernable pathological phenotypes in white-tailed deer with CWD for potential strain differentiation, there is not enough characterization in other cervids to draw conclusions. 

### 4.4. Human Prion Diseases

PrP^Sc^ accumulates in the cerebella of humans with prion diseases [95,163,166,175,176,177]. A summary of these differences can be found in Table 9. The immunostaining pattern in the cerebellar cortex of people with vCJD is described as diffuse and florid plaques [175]. The density of PrP^Sc^ accumulation was increased in the granule cell layer compared to the molecular layer of the cerebellar cortices in people with vCJD [175]. One report on an individual with dura mater-derived iCJD states that there was Type 1 PrP^Sc^ accumulated in the cerebellar cortex. A synaptic staining pattern was found in both the molecular and granule cell layer of the cerebellar cortex [177]. When PrP^Sc^ accumulates in humans with sCJD, reports vary. Some state that people with sCJD accumulate PrP^Sc^ equally in the molecular and granule cell layer of the cerebellum [178], whereas others report that coarse dotted deposits of PrP^Sc^ accumulate in the granule cell layer and diffuse fine dotted deposits in the molecular layer [163,179]. There is evidence that staining patterns vary depending on the subtype of sCJD. People with either the MM1 or MV1 subtype display fine punctate deposits with diffuse aggregates that were occasionally described as coarse [85,89]. PrP^Sc^ plaques in the cerebellum of people with VV1 or MM2 subtype are rare, although more common in people of the MM2 subtype [85,89]. Otherwise, PrP^Sc^ deposition in people of the MM2 subtype is described as coarse [85]. People with the VV2 and MV2 subtype have prominent involvement of the cerebellum [85,89] with these subtypes displaying plaque-like deposits primarily in the granule cell layer [85]. The VV2 subtype does not display Kuru-like deposits while diffuse plaques and punctate deposits are common [89]. In people of the MV2 subtype, kuru-like plaques are prominent [89]. 

Staining patterns are clearly different between people with sCJD compared to people with GSS [163]. Patients with GSS have Kuru plaques with synaptic localization of PrP^Sc^ in both the molecular and granule cell layer [163]. Amyloid deposits of PrP^Sc^ occur in both the molecular and granule cell layers of the cerebellum in people with GSS [95,153] and increased deposition occurs in the molecular layer [165]. For those with Kuru, misfolded prion protein accumulates in the cerebellum [155,157,180]. In the cerebellar cortices of people with Kuru, PrP^Sc^ is not prominent in the molecular layer and is greater in the Purkinje cell and granule cell layer as fine granular diffuse deposits and plaques [155,157,158].

## 5. Enteric Nervous System

In animals with prion diseases, PrP^Sc^ accumulates differentially in the enteric nervous system, a division of the nervous system that spans the entire digestive tract. The differential accumulation of PrP^Sc^ is influenced by route of infection in addition to being both strain-dependent and genotype-dependent. While the enteric nervous system has long been thought to be a portal of PrP^Sc^ entry to the central nervous system, it was not definitively demonstrated until 1999 [181]. 

PrP^Sc^ accumulates differentially in the enteric nervous system of animals with prion diseases and these differences may allow for greater understanding of TSE pathogenesis and strain differentiation. Accumulation is not always predictable by route of infection. Beyond strain, differences in PrP^Sc^ accumulation in the enteric nervous system occur due to propagation differences between natural infection, experimental intracranial inoculation, and experimental oronasal inoculation. Positive accumulation in the enteric nervous system is the result of centrifugal spread throughout the host following oronasal inoculation versus centripetal spread following intracranial inoculation. For example, AA136 sheep that were oronasally inoculated with the scrapie strain x124 were not susceptible to the agent and therefore did not accumulate PrP^Sc^ in the enteric nervous system. However, AA136 sheep that were intracranially inoculated with the x124 scrapie strain were positive for PrP^Sc^ in the enteric nervous system. The No. 13-7 scrapie strain counterparts in this study displayed different results. The No. 13-7 AA136 oronasally inoculated sheep were positive in the enteric nervous system while the intracranially inoculated AA136 sheep were negative [115]. Further, enteric nervous system accumulation did not occur following oral inoculation of atypical L-BSE although the cattle in this study was positive in other peripheral tissues [47]. Interestingly, it has also been reported that peripheral tissues accumulated PrP^Sc^ in cattle intracranially inoculated with BSE; however, the enteric nervous system remained negative [159,182]. In another study, cattle orally challenged with C-BSE demonstrated PrP^Sc^ accumulation in the enteric nervous system throughout the entire time course [183]. In addition to being strain-dependent, the accumulation of misfolded prion protein is genotype-dependent [184]. Here we summarize the differential accumulation of PrP^Sc^ in the myenteric (Auerbach’s) plexus and submucosal (Meissner’s) plexus throughout the enteric nervous system in cattle with BSE, sheep and goats with scrapie, cervids with CWD, and humans with prion diseases (Table 10, Table 11, Table 12, Table 13 and Table 14). We will specify inoculation route throughout.

### 5.1. Bovine Spongiform Encephalopathy 

PrP^Sc^ accumulates in the myenteric plexus and submucosal plexus of the enteric nervous system in cattle with C-BSE following oral inoculation and subsequent PrP^Sc^ uptake [183,185,186,187,188]. A summary of these differences can be found in Table 10. The staining pattern following oral inoculation in the myenteric and submucosal plexuses was not clustered and localized to positive lymph follicles. In these cases, it was suggested that this could be due to direct neuroinvasion that evades gut-associated lymphoid tissue following oral inoculation [186]. The earliest PrP^Sc^ accumulation is observed in the myenteric plexus of cattle with C-BSE is at 16 months post-inoculation in the ileum during preclinical stages [186]. Immunolabelling against PrP^Sc^ is observed to have limited involvement that lasts until clinical stages of the disease following oral inoculation; however, distribution of PrP^Sc^ in the enteric nervous system was wider in cattle with longer incubation periods [186]. Intense staining occurs at end stages of C-BSE in orally inoculated cattle that manifests as linear, intraglial, intraneuronal, and perineuronal with an association to the satellite cell in the myenteric and submucosal plexuses of the enteric nervous system [183,186]. There was no obvious association between immunolabelling against PrP^Sc^ in the enteric nervous system and positive Peyer’s patch follicles [183,186]. H-BSE and L-BSE showed no enteric nervous system involvement, even following the successful oral transmission of atypical BSE [47,159]. Immunohistochemistry failed to demonstrate the presence of PrP^Sc^ accumulation in the enteric nervous system of cattle with E211K H-BSE following intracranial inoculation [53,110].

### 5.2. Scrapie

PrP^Sc^ accumulates differentially in the enteric nervous system of sheep with classical scrapie [181,189,190,191,192], though this is not the case in sheep with atypical scrapie, as these sheep have no PrP^Sc^ immunoreactivity outside the central nervous system [58]. A summary of these differences can be found in Table 11, Table 12 and Table 13. Studies often report the genotype of sheep as a sequence of three one-letter amino acid codes at codons 136, 154, and 171 of the PRNP gene that are of interest due to their influence on susceptibility to scrapie. This section reports genotypes in such a manner. 

The distribution and intensity increase as classical scrapie progresses until the entire enteric nervous system has abundant PrP^Sc^ deposition in clinical stages [189]. In sheep with classical scrapie, intraneuronal fine granules of PrP^Sc^ deposit in the myenteric and submucosal plexuses [181]. Sheep expressing VRQ/VRQ at codons 136, 154, and 171 of the PRNP gene displayed more extensive PrP^Sc^ accumulation in the enteric nervous system than less susceptible sheep. Sheep that expressed VRQ/VRQ were positive in all gastrointestinal sites while sheep that expressed VRQ/ARQ only showed PrP^Sc^ accumulation from the forestomaches to the rectum; sheep that were ARQ/ARQ had PrP^Sc^ accumulation in the enteric nervous system of the omasum, abomasum, and intestines (sans esophagus, reticulum or rumen) [181]. The earliest detection of PrP^Sc^ accumulation in sheep with classical scrapie is at 5 months post inoculation [191]. Sheep with classical scrapie display strong intraneuronal labelling of the ganglion and satellite cells in both myenteric and submucosal plexuses along the large and small intestines [192]. A sparse amount of neurons in the abomasum and duodenum of sheep with classical scrapie had misfolded prion protein accumulation [192]. Unlike cattle, the deposition of PrP^Sc^ in the enteric nervous system of sheep with classical scrapie parallels the extent of deposition in the lymphoreticular system of the gut. Intraneuronal PrP^Sc^ was not in enteric ganglia of forestomaches [192]. PrP^Sc^ accumulation in classical scrapie is genotype-dependent, as enteric neuron staining did not occur in ARR/ARR or ARR/ARQ sheep [192]. This disease-specific intraneuronal immunolabelling of PrP^Sc^ did not occur in the ganglia of the enteric nervous system in the forestomaches of sheep with classical scrapie [192]. With a sensitive protocol, PrP^Sc^ deposition was shown to occur in the ganglia of the myenteric plexus and submucosal plexus in addition to nerve fibers in the submucosa [192]. PrP^Sc^ deposition did not occur in the enteric nervous system near lymphoid nodules associated with inflammatory foci [192] nor was observed in the myenteric plexus at sites apart from large lymphoid aggerates in the jejunum and ileum during points in the incubation period [190]. In the forestomaches of sheep naturally infected with classical scrapie following natural infection, the enteric nervous system of the abomasum is first positive in sheep 9 months old, in which PrP^Sc^ accumulation is present in the autonomous myenteric nervous system (and central nervous system) [190]. This study reports localization near Peyer’s patches [190]. Misfolded prion protein accumulates first in the ileum before progressively spreading to adjacent tissues [190]. In this time course study following natural infection, the most severe case at 9 months of age displayed differential accumulation of PrP^Sc^ in the autonomic myenteric nervous plexus with concomitant accumulation in the medulla oblongata at the level of the obex, specifically in the parasympathetic nucleus of the vagus nerve, that match staining in clinical adult controls [190]. Accumulation of the misfolded prion protein in the autonomic myenteric plexus occurred following its deposition in lymphoid tissues [190]. 

Genotype affects susceptibility to scrapie infection; therefore, there is no PrP^Sc^ accumulation in the enteric nervous system of sheep with resistant genotypes (ARK/ARK and ARQ/ARQ). For example, intracranially inoculated sheep homozygous for lysine at codon 171 of the PRNP gene (ARK/ARK) are negative for PrP^Sc^ in the enteric nervous system, while those with glutamine (ARQ/ARK or ARQ/ARQ) are positive [119]. Additional support for genotype dependence occurs in a study comparing two classical scrapie strains: No. 13-7 and x124 after intracranial (IC) or intranasal (IN) inoculation [115]. Accumulation in sheep with No. 13-7 compared to x124 classical scrapie is both strain-dependent and genotype-dependent in the enteric nervous system. PrP^Sc^ accumulates readily in all genotypes inoculated with No. 13-7, whereas accumulation is less so in sheep with x124 [115]. In sheep with x124 classical scrapie, the most susceptible sheep (VRQ/VRQ) were positive for PrP^Sc^ while the least susceptible (ARQ/ARQ) were negative [115]. Of note, the enteric nervous system was positive in intracranially inoculated sheep with ARQ/ARQ genotype [115]. PrP^Sc^ displayed as fine granules intraneuronally in enteric neurons and glial cells of sheep with classical scrapie as well as the cell membranes of neurons of both the myenteric and submucosal plexuses [115]. Immunolabelling of the foregut varied by strain and genotype [115]. The enteric nervous system of the reticulum had PrP^Sc^ deposits (13-7 IC ARQ/ARQ), rumen and abomasum (13-7 IN VRQ/VRQ), and omasum (x124 IN VRQ/VRQ, 13-7 IN ARQ/VRQ) [115]. The PrP^Sc^ deposition varied in the enteric nervous system of the jejunum, ileum, and cecum [115]. Immunolabelling in the enteric nervous system occurs at the same time as gut-associated lymphoid tissue in sheep with No. 13-7 classical scrapie, whereas concurrent accumulation in sheep with x124 only occurred in VRQ/VRQ sheep [115]. Interestingly, the enteric nervous system did not show PrP^Sc^ accumulation in ARQ/ARQ sheep with oronasally inoculated x124, who were not susceptible to scrapie, while it was positive in ARQ/ARQ sheep with intracranially inoculated x124 [115].

The onset of PrP^Sc^ accumulation in the enteric nervous system is genotype dependent in sheep with classical scrapie [184]. Sheep expressing VRQ/VRQ in the PRNP gene the enteric nervous system was positive in all sheep at 112 dpi [184]. The following is a list of genotypes in increasing dpi at onset of PrP^Sc^ accumulation in the enteric nervous system in sheep with classical scrapie: VRQ/ARQ (177 dpi), ARQ/ARQ (220 dpi), VRQ/ARR (366 dpi) [184]. The longest time period it took for any sheep in a genotype cohort to present with PrP^Sc^ in the enteric nervous system was 2252 dpi in sheep that are ARQ/ARR [184]. Overall, this indicates that sheep more susceptible genotypes accumulate PrP^Sc^ earlier in the enteric nervous system than more resistant sheep [184]. 

Further, intraneuronal accumulation occurred in enteric glial cells [193]. Sheep homozygous for ARR/ARR one month following inoculation with classical scrapie did not show PrP^Sc^ accumulation and other sheep in this cohort remained heathy at 1.5 years following inoculation [194]. Sheep with clinical classical scrapie displayed widespread PrP^Sc^ deposition in the enteric nervous system [194]. In this study, only sheep with clinical scrapie had PrP^Sc^ accumulation in the enteric nervous system while this did not occur in sheep at earlier time points. Some studies suggest that amplification in the lymphoreticular system occurs prior to accumulation of PrP^Sc^ in the enteric nervous system of sheep with classical scrapie [189,191] while others suggest simultaneous exposure to infection [194]. In sheep with atypical scrapie, there is no deposition of PrP^Sc^ in the enteric nervous system regardless of genotype [58,195]. In other reports of atypical scrapie in sheep, involvement of the enteric nervous system was not investigated [3,196]. In goats with classical scrapie, PrP^Sc^ is reported in the myenteric and submucosal plexuses of the enteric nervous system [122,123]. 

### 5.3. Chronic Wasting Disease

PrP^Sc^ accumulates in the myenteric and submucosal plexuses of the enteric nervous system in cervids with CWD [129,131,133,197,198]. A summary of these differences can be found in Table 14. In white-tailed deer that were orally inoculated with CWD, PrP^Sc^ accumulation occurred in the nerve fibers and ganglia of the enteric nervous system throughout the intestine [129]. Distribution of PrP^Sc^ accumulation was genotype-dependent in white-tailed deer with CWD [129]. White-tailed deer with the genotypes H95/G96 or H95/S96 had less PrP^Sc^ accumulation in the villi and crypts of the intestinal mucosa than white-tailed deer with the genotypes Q95/G96 or Q95/S96 deer [129]. This difference happened to the greatest degree at the ileocecal junction, as Q95/G96 and Q95/S96 white-tailed deer displayed strong PrP^Sc^ accumulation, whereas H95/G96 and H95/S96 deer showed no PrP^Sc^ accumulation [129]. In Rocky Mountain elk naturally infected with CWD, PrP^Sc^ accumulation in the enteric nervous system was evident in the myenteric and submucosal plexuses [197]. In reindeer orally inoculated with CWD, PrP^Sc^ accumulation is prominent in the myenteric and submucosal plexuses throughout the intestines [131]. In mule deer with CWD, reports vary and are limited. One report states that PrP^Sc^ accumulation did not occur in the myenteric plexus following natural infection [121], while another report on PrP^Sc^ in the enteric nervous system of mule deer states it is positive following natural infection [198].

### 5.4. Human Prion Diseases

There is evidence of expression of PrP^C^ in the human enteric system [199]. However, there is no data to support PrP^Sc^ accumulation in the enteric nervous system in people with CJD or GSS [200]. In a largescale survey of appendectomies in Britain, misfolded prion protein was found to be in the appendix of individuals, a handful of cases resulted in positive immunolabelling for PrP^Sc^ in the appendix [201]. As the appendix in humans contains enteric neural tissue [202], future studies are necessary to investigate and characterize potential PrP^Sc^ accumulation in the enteric nervous system of people with vCJD. PrP^Sc^ accumulation is not investigated in cases of Kuru.

## 6. Conclusions

This review focused on three regions of the nervous system where neuroinvasion and subsequent PrP^Sc^ accumulation is often different, in different strains of TSE within a given species. The retinal ganglion cells of the retina, the cerebellar cortex, and the enteric neurons of the foregut are three regions that may be useful to examine as additional TSE strains emerge and are needed to be characterized in the natural host. 

## Figures and Tables

**Figure 1 viruses-13-02453-f001:**
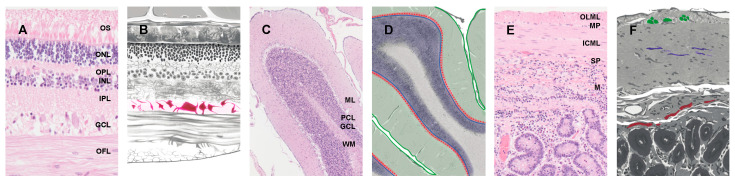
Retinal ganglion cells, cerebellar cortex and white matter, and plexuses of the enteric nervous system from animals with TSEs: (**A**) normal retina stained with hematoxylin and eosin. OS (outer segments of photoreceptor cells), ONL (outer nuclear layer), OPL (outer plexiform layer), INL (inner nuclear layer), IPL (inner plexiform layer), GCL (ganglion cell layer), OFL (optic fiber layer). Original magnification 20× (**B**) illustrated retina with the retinal ganglion cells highlighted in red and ganglion cell layer highlighted in pink; (**C**) normal cerebellum stained with hematoxylin and eosin. ML (molecular layer), PCL (Purkinje cell layer), GCL (granule cell layer), WM (white matter). Original magnification 5×; (**D**) illustrated cerebellum with the molecular layer outlined and highlighted in green, the Purkinje cell layer lined with red dots, the granule cell layer highlighted in purple, and the white matter highlighted in gray; (**E**) normal gut cross-section stained with hematoxylin and eosin. OLML (outer longitudinal muscle layer), MP (myenteric plexus), ICML (inner circular muscular layer), SP (submucosal plexus), M (mucosa). Original magnification 10×; (**F**) illustrated gut cross-section with the myenteric plexus highlighted in green, nerve fibers highlighted in purple, and submucosal plexus highlighted in red. Each image is original and previously unpublished.

**Figure 2 viruses-13-02453-f002:**
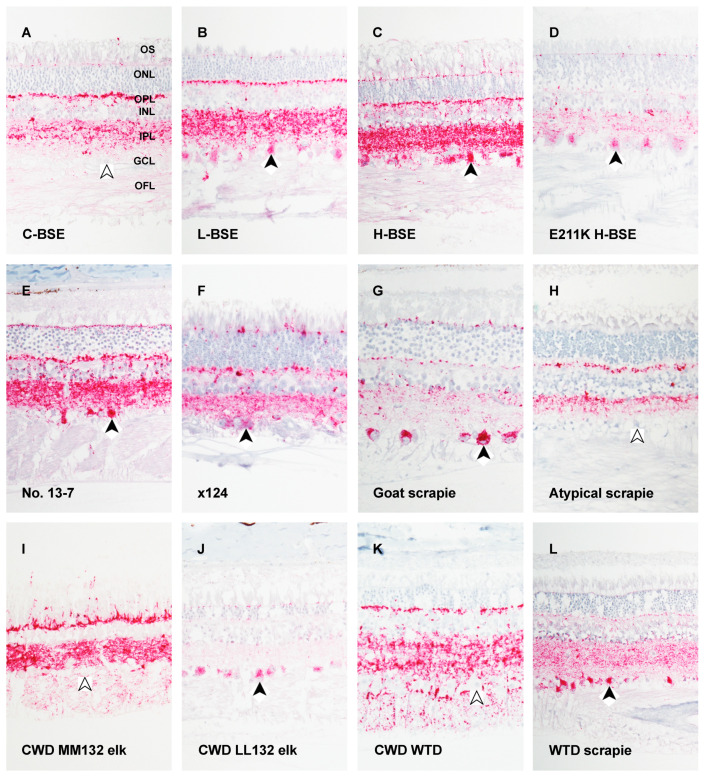
Strain-dependent differential accumulation of PrP^Sc^ in the retinas of animals with TSEs: (**A**–**D**) retinas of cattle with BSE. Cattle with L-BSE, H-BSE, and E211K H-BSE have intraneuronal staining of PrP^Sc^ in the cell body of the retinal ganglion cells (solid arrows) while cattle with C-BSE do not (open arrow). C-BSE (classical bovine spongiform encephalopathy), L-BSE (low-type bovine spongiform encephalopathy), H-BSE (high-type bovine spongiform encephalopathy), E211K H-BSE (high-type bovine spongiform encephalopathy with EK polymorphism at codon 211 of PRNP gene); (**E**–**H**) retinas of sheep and goats with classical and atypical scrapie. Sheep and goats with classical scrapie strains have immunoreactivity against PrP^Sc^ in the cell body of retinal ganglion cells (solid arrows). Sheep with atypical scrapie do not accumulate PrP^Sc^ in the cell body of retinal ganglion cells (open arrow). No. 13-7 (classical scrapie strain in sheep), x124 (classical scrapie strain in sheep), goat scrapie (classical scrapie in goat), atypical scrapie (atypical scrapie in sheep); (**I**–**L**) retinas from Rocky Mountain elk with CWD and white-tailed deer with CWD or scrapie. Rocky Mountain elk that are of the LL132 genotype have PrP^Sc^ accumulation in the cell body of retinal ganglion cells (solid arrow) while those expressing MM132 do not (open arrow). White-tailed deer with CWD does not accumulate PrP^Sc^ in the retinal ganglion cells (open arrow), while white-tailed deer experimentally inoculated with scrapie do accumulate PrP^Sc^ in retinal ganglion cells (solid arrow). CWD MM132 elk (chronic wasting disease in elk expressing homozygous methionine at codon 132 of the PRNP gene), CWDLL132 elk (chronic wasting disease in elk expressing homozygous leucine at codon 132 of the PRNP gene), CWD WTD (chronic wasting disease in white-tailed deer), WTD scrapie (white-tailed deer inoculated with classical scrapie). Original magnification 20×. Each image is original and previously unpublished.

**Table 1 viruses-13-02453-t001:** Immunolabelling patterns of PrP^Sc^ in the retinal ganglion cells in cattle with BSE.

Strain	Genotype	Immunolabelling Pattern
Classical		Rarely affected (West Greenlee 2015) … decreased/much less prominent than E211K H-BSE (Moore 2016)
H-type		Large globular deposits (West Greenlee 2015) … robust intracellular deposits, increased compared to C-BSE (Mammadova 2020) … intense granular in ganglion cell layer (Okada 2011)
H-type	E211K	Intense (Greenlee 2012) … Increased accumulation compared to BSE-C, prominent intraneuronal (Moore 2016)
L-type		Increased compared to BSE-C (Mammadova 2020) … intracellular, intense punctate (Smith 2009)

**Table 2 viruses-13-02453-t002:** Immunolabelling patterns of PrP^Sc^ in the retinal ganglion cells in sheep and goats with scrapie.

Species	Strain	Genotype	Immunolabelling Pattern
Sheep	Classical (intracranial)	AA136	Granular and coarse particulate deposits (Jeffrey 2014)
	Classical (field)		Granular (Regnier 2011)
	Classical (No. 13-7)	VV136	Multifocal punctate in ganglion cell layer (Moore 2016)
		AV136	Multifocal punctate in ganglion cell layer (Moore 2016) … intense (Smith 2008)
		AA136	Multifocal punctate in ganglion cell layer (Moore 2016) … intense (Smith 2008)
		KK171	-
		QK171	-
		QQ171	-
	Classical (x124)	VV136	Multifocal punctate in ganglion cell layer (Hamir 2009, Moore 2016)
		AV136	Multifocal punctate in ganglion cell layer (Hamir 2009, Moore 2016)
		AA136	Multifocal punctate in ganglion cell layer (Hamir 2009, Moore 2016 *)
	Atypical	VRQ/ARQ	Minimal in ganglion cell layer (Cassmann 2021)
		ARQ/ARQ	Minimal in ganglion cell layer (Cassmann 2021)
		ARQ/ARR	Minimal in ganglion cell layer (Cassmann 2021)
		AHQ/ARQ	-
		AHQ/AHQ	-
		ARR/ARR	-
Goat	Classical		Marked intraneuronal (Mammadova 2020) … strong prominent cytoplasmic (Valdez 2003)

(-) Not reported. * Intranasally inoculated were negative. Intracranially inoculated were positive.

**Table 3 viruses-13-02453-t003:** Immunolabelling patterns of PrP^Sc^ in the retinal ganglion cells in cervids with CWD.

Species	Genotype	Immunolabelling Pattern
RME	132MM	Absence of intracytoplasmic staining (Spraker 2010)
	132ML	Intracytoplasmic staining present (Spraker 2010)
	132LL	Prominent, heavy intracytoplasmic staining compared to ML (Spraker 2010)
RD		Intraneuronal (Moore 2016)
WTD	Q95/G96	Rare intraneuronal staining (Lambert, unpublished)
	Q95/S96	Rare intraneuronal staining (Lambert, unpublished)
	H95/G96	-
	H95/S96	*-*
MD		Present in ganglion cell layer (Spraker 2002)

(-) Not reported.

**Table 4 viruses-13-02453-t004:** Immunolabelling patterns of PrP^Sc^ in the retinal ganglion cells in humans with CJD.

Strain	Subtype	Immunolabelling Pattern
sCJD	MM1	No intraneuronal staining (Head 2003, Takao 2018)
	MV1	-
	VV1	-
	MM2	No intraneuronal staining (Head 2003, Takao 2018)
	MV2	No intraneuronal staining (Head 2003, Takao 2018)
	VV2	-
iCJD		-
vCJD		No intraneuronal staining (Head 2003, Takao 2018)

(-) Not reported.

**Table 5 viruses-13-02453-t005:** Immunolabelling patterns of PrP^Sc^ in the cerebellar cortex in cattle with BSE.

Strain	Genotype	Relative Accumulation	Molecular Layer	Purkinje Cell Layer	Granule Cell Layer
Classical		Lower than brainstem at level of obex (Polak 2012, Vidal 2006)	Prominent stellate and linear (Moore 2016) … moderate stellate; apparent perineuronal, fine punctate, coarse particulate; barely apparent intraneuronal (Siso 2004) … moderate or high stellate (Okada 2011)	-	Fine and coarse granular to aggregated (Moore 2016)
High-type		Lower than brainstem at level of obex (Polak 2012)	Minimal and less uniformly distributed compared to BSE-L (Konold 2012) … prominent stellate labeling (Okada 2011)	-	Minimal and less uniformly distributed compared to BSE-L (Konold 2012) … prominent stellate labeling (Okada 2011),
High-type	E211K	-	Fine granular and stellate (Moore 2016) … small, multifocal clumps of granular and particulate staining (Greenlee 2012)	-	coarse granular (Moore 2016) … scant; small, multifocal clumps of granular and particulate staining (Greenlee 2012)
Low-type		Equal to brainstem at level of obex (Polak 2012, Okada 2017)	Very homogenous involvement, diffuse and even (Konold 2012)	-	Very homogenous involvement, diffuse and even (Konold 2012)

(-) Not reported.

**Table 6 viruses-13-02453-t006:** Immunolabelling patterns of PrP^Sc^ in the cerebellar cortex in sheep and goats with classical scrapie.

Species	Strain	Genotype	Relative Accumulation	Molecular Layer	Purkinje Cell Layer	Granule Cell Layer
Sheep	Classical (No. 13-7)	VV136	-	-	-	-
		AV136	-	Intense, multifocal (Greenlee 2019)	-	Intense, stronger than molecular layer (Greenlee 2019)
		AA136	-	-	-	-
		KK171	-	Scant punctate, granular, and stellate deposits (Cassmann 2019)	Small amount of intraneuronal (Cassmann 2019)	Moderate but variable amounts of granular, intraneuronal, and intraglial (Cassmann 2019)
		QK171	-	Scant punctate, granular, and stellate deposits (Cassmann 2019)	Small amount of intraneuronal (Cassmann 2019)	Moderate but variable amounts of granular, intraneuronal, and intraglial (Cassmann 2019)
		QQ171	-	Scant punctate, granular, and stellate deposits (Cassmann 2019)	Small amount of intraneuronal (Cassmann 2019)	Moderate but variable amounts of granular, intraneuronal, and intraglial (Cassmann 2019)
	Classical (x124)	VV136	-	-	-	-
		AV136	-	-	-	-
		AA136	-	-	-	-
Goat	Classical		-	Subpial (Mammadova 2020) … strong and widespread cytoplasmic immunoreactivity of small neurons (Valdez 2003)	Strong and widespread cytoplasmic immunoreactivity of Purkinje cells (Valdez 2003)	Strong and widespread cytoplasmic immunoreactivity of small neurons (Valdez 2003)

(-) not reported.

**Table 7 viruses-13-02453-t007:** Immunolabelling patterns of PrP^Sc^ in the cerebellar cortex in sheep with atypical scrapie.

Genotype	Relative Accumulation	Molecular Layer	Purkinje Cell Layer	Granule Cell Layer
VRQ/ARQ	-	Granular and punctate (Cassmann 2021)…decreased severity relative to granule cell layer (Nentwig 2007),	-	Increased severity relative to molecular layer (Nentwig 2007)
ARQ/ARQ	Higher than brainstem at level of obex (Moore 2008)	fine granular, linear, aggregates (Moore 2008) … granular and punctate (Cassmann 2021) … fine punctate to coarse granular restricted to neuropil of subpial molecular layer (Okada 2016) … similar severity to granule cell layer (Nentwig 2007),	Always negative intraneuronally (Moore 2008)	Moderate to marked fine granular (Moore 2008)…similar severity to molecular layer (Nentwig 2007)
ARQ/ARR	-	intense, stronger than granule cell layer (Greenlee 2019) … granular and punctate (Cassmann 2021)	-	Intense (Greenlee 2019)
AHQ/ARQ	Higher than brainstem at level of obex (Benestad 2003, Moore 2008)	Fine granular, linear, aggregates (Moore 2008) … intense and marked synaptic (Benestad 2003) … similar severity to granule cell layer (Nentwig 2007)	Always negative intraneuronally (Moore 2008)	Moderate to marked fine granular (Moore 2008) … pronounced and widspread (Benestad 2003) … similar severity to molecular layer (Nentwig 2007)
AHQ/AHQ	Higher than brainstem at level of obex (Benestad 2003, Moore 2008)	Fine granular, linear, aggregates (Moore 2008) … intense and marked synaptic (Benestad 2003)	Always negative intraneuronally (Moore 2008)	Moderate to marked fine granular (Moore 2008) … pronounced and widspread (Benestad 2003)
ARR/AHQ	Higher than brainstem at level of obex (Moore 2008)	Fine granular, linear, aggregates (Moore 2008)	Always negative intraneuronally (Moore 2008)	Moderate to marked fine granular (Moore 2008)
ARR/ARR	Higher than brainstem at level of obex (Kittelberger 2010)	Fine granular, linear, aggregates (Moore 2008) … increased severity relative granule cell layer (Nentwig 2007)	Always negative intraneuronally (Moore 2008)	Moderate to marked fine granular (Moore 2008) … decreased severity relative to molecular layer (Nentwig 2007)

(-) Not reported.

**Table 8 viruses-13-02453-t008:** Immunolabelling patterns of PrP^Sc^ in the cerebellar cortex in cervids with CWD.

Species	Genotype	Relative Accumulation	Molecular Layer	Purkinje Cell Layer	Granule Cell Layer
RME	132MM	-	-	-	-
	132ML	-	-	-	-
	132LL	-	-	-	-
RD		Lower than brainstem at level of obex (Benestad 2016)	Punctate, particulate, aggregated deposits (Moore 2016) … moderate stellate (Benestad 2016)	Absence of intraneuronal staining (Moore 2016)	Punctate, particulate (Moore 2016) … patchy, heavier than molecular layer (Benestad 2016)
WTD	Q95/G96	-		Plaques extending from granule cell layer (Otero 2019)	Severe, coarse granular and large plaques extending to molecular layer (Otero 2019)
	Q95/S96	-	Milder granular and diffuse compared to Q95/G96 (Otero 2019)	Milder granular and diffuse compared to Q95/G96 (Otero 2019)	Plaques restricted here (Otero 2019)
	H95/G96	-	-	-	Discontinuous and diffuse, fine punctate and coarse small granular, a few plaque-like deposits (Otero 2019)
	H95/S96	-	More intense immunolabeling than other genotypes, conspicuous stellate aggregates (Otero 2019)	-	Fine punctate and coarse granular deposits homogeneously distributed (Otero 2019)
MD		-	Amyloid plaques (Guiroy 1991)	-	Amyloid plaques, immunoreactive material also in neuronal perikarya (Guiroy 1991)

(-) Not reported.

**Table 9 viruses-13-02453-t009:** Immunolabelling patterns of PrPSc in the cerebellar cortex in humans with prion disease.

Strain	Subtype	Relative Accumulation	Molecular Layer	Purkinje Cell Layer	Granule Cell Layer
sCJD	NA	-	Diffuse and fine dotted deposits, synatpic type (Yang 1999) … diffuse irregular plaque-like deposits, punctate synaptic-like deposits, fine punctate (Ferrer 2000)	No intraneuronal staining (Ferrer 2000)	Coarse dotted deposits (Yang 1999) … diffuse irregular plaque-like deposits, punctate synaptic-like deposits, aggregated granules in midst of somas of granular cells (Ferrer 2000)
	MM1	Higher than brainstem at level of obex (Parchi 1996)	Cerebellar synaptic (Parchi 1999)	-	Cerebellar synaptic (Parchi 1999)
	MV1	Higher than brainstem at level of obex (Parchi 1996)	Cerebellar synaptic (Parchi 1999)	-	Cerebellar synaptic (Parchi 1999) … plaque-like deposits (Parchi 1996)
	VV1	Higher than brainstem at level of obex (Parchi 1996)	Cerebellar synaptic (Parchi 1999) … intense and widespread punctate (Parchi 1996)	-	Cerebellar synaptic (Parchi 1999) … intense and widespread punctate, plaque-like pattern (Parchi 1996)
	MM2	Less than or equal to brainstem at level of obex (Parchi 1996)	Cerebellar synaptic (Parchi 1999)	-	Cerebellar synaptic (Parchi 1999) … plaque-like deposits (Parchi 1996)
	MV2	Higher than brainstem at level of obex (Parchi 1996)	Cerebellar synaptic (Parchi 1999)	-	Plaque-like (Parchi 1996) … cerebellar synaptic (Parchi 1999)
	VV2	Higher than brainstem at level of obex (Parchi 1996)	Cerebellar synaptic (Parchi 1999) …… intense and widespread punctate (Parchi 1996)	-	Cerebellar synaptic (Parchi 1999) … intense and widespread punctate, plaque-like pattern (Parchi 1996)
iCJD		-	Synaptic (Shijo 2017)	-	Synaptic (Shijo 2017)
vCJD		-	Diffuse and florid plaques (Armstrong 2009)	No staining evident (Armstrong 2009)	Diffuse and florid plaques, greater density than molecular layer (Armstrong 2009)
GSS		Lower than brainstem at level of obex (Bugiani 2000)	Kuru plaques, synaptic-type depositions (Yang 1999) … large multicore plaques (Budka 1995) … amyloid deposits (Bugiani 2000)	-	Kuru plaques, synaptic-type depositions (Yang 1999) … fewer and smaller kuru plaques (Budka 1995) … amyloid plaques (Bugiani 2000)
KURU		-	Occasional kuru plaques (Brandner 2008) … synaptic type and plaque deposits, florid plaques (Hainfellner 1997)	Plaques (Hainfellner 1997, Bradner 2008)	Occasional kuru plaques (Brandner 2008) … Kuru plaques most numerous here (Liberski 2012) … synaptic type and plaque deposits, more prominent than in molecular layer (Hainfellner 1997)

(-) Not reported.

**Table 10 viruses-13-02453-t010:** Immunolabelling patterns of PrP^Sc^ in the enteric nervous system in cattle with BSE.

Strain	Genotype	ENS	Submucosal Plexus	Myenteric Plexus
Classical		Positive (Balkema-Buschmann 2011, Hoffmann 2011, Kaatz 2012, Franz 2012) … no immunolabelling (Konold 2012, Moore 2016)	Perineuronal and an association to satellite cells, intraneuronal, intraglial (Kaatz 2012)	Perineuronal and an association to satellite cells, intraneuronal, intraglial (Kaatz 2012) … sparse immmunostaining, fine granular deposits (Terry 2003)
H-type		No immunolabelling (Konold 2012)	-	-
H-type	E211K	Negative (Greenlee 2012, Moore 2016)	-	-
L-type		Negative (Okada 2017) … no immunolabelling (Konold 2012)	-	-

(-) Not reported.

**Table 11 viruses-13-02453-t011:** Immunolabelling patterns of PrP^Sc^ in the enteric nervous system in sheep with naturally occurring classical scrapie.

Genotype	ENS	Submucosal Plexus	Myenteric Plexus
VRQ/VRQ	Positive (van Keulen 1999, 2002; Andreoletti 2000, Gonzalez 2014)	Positive (van Keulen 1999, 2002) … intraneuronal (Andreoletti 2000)	Positive (van Keulen 1999, 2002)
VRQ/ARQ	Positive (van Keulen 1999, Gonzalez 2014)	-	-
VRQ/ARH	Positive (van Keulen 1999)	-	-
VRQ/AHQ	Positive (van Keulen 1999)	-	-
ARQ/ARQ	Positive (van Keulen 1999, Heggebo 2003, Gonzalez 2014, Marruchella 2007, Jeffrey 2006)	Intraneuronal (Heggebo 2003, Marruchella 2007) … granular (Marruchella 2007)	Intraneuronal (Heggebo 2003, Marruchella 2007) … granular (Marruchella 2007)
VRQ/ARR	Positive (Gonzalez 2014)	-	-
ARQ/ARR	Positive (Gonzalez 2014, Marruchella 2007)	Granular, intraneuronal (Marruchella 2007)	Granular, intraneuronal (Marruchella 2007)
ARR/ARR	Positive (van Keulen 1999)…clinically normal (Andreoletti 2000, Jeffrey 2006)	Granular, intraneuronal (Marruchella 2007)	Granular, intraneuronal (Marruchella 2007)

(-) Not reported.

**Table 12 viruses-13-02453-t012:** Immunolabelling patterns of PrP^Sc^ in the enteric nervous system in sheep and goats experimentally inoculated with classical scrapie.

Species	Strain	Genotype	ENS	Submucosal Plexus	Myenteric Plexus
Sheep	Classical (No. 13-7)	VV136	Intraneuronal and intraglial fine granules (Moore 2016)	Intraneuronal and intraglial fine granules, occassionally cell membranes (Moore 2016)	Intraneuronal and intraglial fine granules, occasional cell membranes (Moore 2016)
		AV136	Intraneuronal and intraglial fine granules (Moore 2016)	Intraneuronal and intraglial fine granules, occassionally cell membranes (Moore 2016)	Intraneuronal and intraglial fine granules, occasional cell membranes (Moore 2016)
		AA136	Intraneuronal and intraglial fine granules (Moore 2016)	Intraneuronal and intraglial fine granules, occassionally cell membranes (Moore 2016)	Intraneuronal and intraglial fine granules, occasional cell membranes (Moore 2016)
		KK171	Negative (Cassmann 2019)	Negative (Cassmann 2019)	Negative (Cassmann 2019)
		QK171	Positive (Cassmann 2019)	Positive (Cassmann 2019)	Positive (Cassmann 2019)
		QQ171	Positive (Cassmann 2019)	Positive (Cassmann 2019)	Positive (Cassmann 2019)
	Classical (x124)	VV136	Intraneuronal and intraglial fine granules (Moore 2016)	Intraneuronal and intraglial fine granules, occassionally cell membranes (Moore 2016)	Intraneuronal and intraglial fine granules, occasional cell membranes (Moore 2016)
		AV136	Intraneuronal and intraglial fine granules (Moore 2016)	Intraneuronal and intraglial fine granules, occassionally cell membranes (Moore 2016)	Intraneuronal and intraglial fine granules, occasional cell membranes (Moore 2016)
		AA136	Dependent on inoculation route (Moore 2016 *)	Dependent on inoculation route (Moore 2016 *)	Dependent on inoculation route (Moore 2016 *)
Goat	Classical		Positive (Valdez 2003, Mammadova 2020)	Positive (Valdez 2003)	Positive (Valdez 2003)

(-) Not reported. * Intranasally inoculated sheep were negative. Intracranially inoculated sheep were positive.

**Table 13 viruses-13-02453-t013:** Immunolabelling patterns of PrP^Sc^ in the enteric nervous system in sheep experimentally inoculated with atypical scrapie.

Genotype	ENS	Submucosal Plexus	Myenteric Plexus
VRQ/ARQ	Negative (Cassmann 2021)	Negative (Cassmann 2021)	Negative (Cassmann 2021)
ARQ/ARQ	Negative (Cassmann 2021)	Negative (Cassmann 2021)	Negative (Cassmann 2021)
ARQ/ARR	Negative (Cassmann 2021)	Negative (Cassmann 2021)	Negative (Cassmann 2021)
AHQ/ARQ	-	-	-
AHQ/AHQ	-	-	-
ARR/ARR	-	-	-

(-) Not reported.

**Table 14 viruses-13-02453-t014:** Immunolabelling patterns of PrP^Sc^ in the enteric nervous system in cervids with CWD.

Species	Genotype	ENS	Submucosal Plexus	Myenteric Plexus
RME	132MM	Positive (Spraker 2009)	On periphery of enteric neurons, nonmyelinated nerves located within the intestinal submucosa (Spraker 2009)	Granules surrounding myenteric neurons located in ganglion between bundles of smooth muscle (Spraker 2009)
	132ML	Positive (Spraker 2009)	On periphery of enteric neurons, nonmyelinated nerves located within the intestinal submucosa (Spraker 2009)	Granules surrounding myenteric neurons located in ganglion between bundles of smooth muscle (Spraker 2009)
	132LL	-	-	-
RD		Prominent (Mitchell 2012)	Granular (Mitchell 2012)	Granular (Mitchell 2012)
WTD	Q95/G96	Positive (Otero 2019, Mammadova 2020)	-	-
	Q95/S96	Positive (Otero 2019, Mammadova 2020)	-	-
	H95/G96	Positive (Otero 2019)	-	-
	H95/S96	Positive (Otero 2019)	-	-
MD		Not found (Spraker 2002)…positive, coarse (Sigurdson 2001)	-	Not found (Spraker 2002) … positive, coarse stain, in myenteric ganglion cell bodies, along nerve fibers and in satellite cells (Sigurdson 2001)

(-) Not reported.

## Data Availability

Not applicable.

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
