# Peer review of "Differential Accumulation of Misfolded Prion Strains in Natural Hosts of Prion Diseases"

_viruses, 2021, doi:10.3390/v13122453_

Round 1

Reviewer 1 Report

The review by Lambert et al. is well written  and provides a comprehensive view of how PrPsc accumulates differently in three different regions of the nervous system according to the prion strain and natural host.

I only have a few minor issues that need to be considered before the paper can be accepted for publication:

  • I found the title to be a little long and complicated. I suggest to change to "Differential accumulation of misfolded prion strains in natural hosts of prion diseases"
  • The authors could show a few examples of PrP western blots representative of the cases they discuss.
  • The authors could try to better explain in the introduction why they chose to focus on the retinal ganglion cells, cerebellar cortex and enteric nervous system.
  • The authors should provide references for the images they show in Figures 1 and 2.

Author Response

The authors appreciate the thoughtful review, and the opportunity to improve this manuscript. Our responses to reviewer comments are detailed below. Solid bullets indicate reviewer comment, and open bullets indicate changes made to the manuscript.

  • I found the title to be a little long and complicated. I suggest to change to "Differential accumulation of misfolded prion strains in natural hosts of prion diseases"
    • The title is now "Differential accumulation of misfolded prion strains in natural hosts of prion diseases"

  • The authors could show a few examples of PrP western blots representative of the cases they discuss.
    • Rather than create new western blots, the authors now reference “The Transmissible Spongiform Encephalopathies of Livestock” by Greenlee et al. This citation includes a western blot with samples from C-BSE, H-BSE, and L-BSE that demonstrate different banding patterns.
    • Line 52-53

  • The authors could try to better explain in the introduction why they chose to focus on the retinal ganglion cells, cerebellar cortex and enteric nervous system.
    • Previous: In this review we focus on PrPSc accumulation in three locations in the nervous system (retinal ganglion cells, cerebellar cortex, and enteric nervous system; see Figure 1) and discuss their utility in differentiating strains and better understanding the pathogenesis of BSE in cattle, scrapie in sheep and goats, and chronic wasting disease in cervids.
    • Revision: In this review we focus on PrPSc accumulation in three locations in the nervous system (retinal ganglion cells, cerebellar cortex, and enteric nervous system; see Figure 1). These three nervous system sites were selected based on the presence of robust literature that reports strain-dependent differential accumulation of PrPSc within these structures. We discuss their utility in differentiating strains and better understanding the pathogenesis of bovine spongiform encephalopathy in cattle, scrapie in sheep and goats, and chronic wasting disease in cervids.
    • Lines 71-77

  • The authors should provide references for the images they show in Figures 1 and 2.
    • Each image is original and previously unpublished.
    • This is now stated at the end of each figure legend
    • Lines 90 and 302

Reviewer 2 Report

November 23, 2021

Comments to the Editor and authors:

As far as I confirmed the references of this review, Lambert et al. correctly compiled data on human and animal prion diseases, especially for the localization of PrPSc in three locations in the nervous system. Authors discussed utility of these tissues in differentiating prion strains and the pathogenesis of animal and human prions in their natural hosts based on a large body of experimental evidence. The manuscript is basically well written, and I found their work of interest. However, there are some issues that should be addressed before publication.

Minor points

  1. Line 54-61: Authors state that animals with classical cases shed prions into the environment, on the other hand, atypical cases shed little to no prions into the environment. In case of scrapie, it is correct. However, this classification cannot apply to BSE because both classical and atypical BSE cases shed no BSE prions into the environment. Author’s description causes misunderstanding to the readers.
  2. Line 91: REF 38 is not appropriate for the reference to C-BSE transmission to cattle. It seems to be the reference to C-BSE transmission to humans.
  3. Line 121-123: It might be correct. But at least two reports indicate the possibility that scrapie prions can transmit to humans as described below. Therefore, it is fair to the readers to mention both possibilities in the review.

Cassard H, Torres JM, Lacroux C, Douet JY, Benestad SL, Lantier F, et al. Evidence for zoonotic potential of ovine scrapie prions. Nat Commun. 2014;5:5821. doi: 10.1038/ncomms6821.

Comoy EE, Mikol J, Luccantoni-Freire S, Correia E, Lescoutra-Etchegaray N, Durand V, et al. Transmission of scrapie prions to primate after an extended silent incubation period. Scientific reports. 2015;5:11573. doi: 10.1038/srep11573.

  1. Line 517-518: Authors described that the withe-tailed deer wild-type prion protein allele is Q95/S96… I think that Q95/G96 is correct.
  2. RME in Table 2.3: Authors mentioned the distributions and PrPSc types in the cerebellar cortex of Rocky Mountain elk affected with CWD, and the data were from author’s lab. Therefore, RME in Table 2.3 should be filled out. Otherwise, tell me the reason why (-) not reported.
  3. iCJD in Table 2.4: At least, one report states that immunohistochemistry for PrP (3F4) showed a synaptic staining pattern in both the molecular and granular layers in cerebellar cortex of Dura mater graft-associated CJD patient (Shijo et al. 2017. Neuropathology). 
  4. Line 768: REF 198 is not appropriate. For example, Gill et al. 2013. BMJ may be an appropriate reference.
  5. Line 1098: The title of REF 128 is something wrong. Please reconfirm the reference.

Author Response

The authors appreciate the thoughtful review, and the opportunity to improve this manuscript. Our responses to reviewer comments are detailed below. Solid bullets indicate reviewer comment, and open bullets indicate changes made to the manuscript.

  • Line 54-61: Authors state that animals with classical cases shed prions into the environment, on the other hand, atypical cases shed little to no prions into the environment. In case of scrapie, it is correct. However, this classification cannot apply to BSE because both classical and atypical BSE cases shed no BSE prions into the environment. Author’s description causes misunderstanding to the readers.
    • This is now clarified.
    • Previous: Strains also may be grouped into what are referred to as ‘classical’ and ‘atypical’ based on proteinase K-resistant fragments, neuronal tropism, deposition patterns, and pathological lesion profile [29]. Classical cases of prion diseases tend to occur in groups of younger animals compared to atypical cases that tend to occur in individual older animals. Additionally, animals with classical cases shed prions into the environment and thereby have a propensity for vertical and horizontal transmission under field conditions. Atypical cases of prion diseases shed little to no prions into the environment, providing further evidence to support the spontaneous origin of atypical prion diseases.
    • Revision: Strains also may be grouped into what are referred to as ‘classical’ and ‘atypical’ based on proteinase K-resistant fragments, neuronal tropism, deposition patterns, and pathological lesion profile [30]. Classical cases of prion diseases tend to occur in groups of younger animals compared to atypical cases that tend to occur in individual older animals. Additionally, animals with classical cases of scrapie shed prions into the environment and thereby have a propensity for vertical and horizontal transmission under field conditions. Atypical cases of scrapie shed little to no prions into the environment, providing further evidence to support the spontaneous origin of atypical prion diseases. This, however, does not hold true in cases of cattle in which prions are not shed into the environment regardless of strain.
    • Lines 54-63

  • Line 91: REF 38 is not appropriate for the reference to C-BSE transmission to cattle. It seems to be the reference to C-BSE transmission to humans.
    • This reference has been removed from cattle and is now cited for humans.
    • Line 97

  • Line 121-123: It might be correct. But at least two reports indicate the possibility that scrapie prions can transmit to humans as described below. Therefore, it is fair to the readers to mention both possibilities in the review.
  1. Cassard H, Torres JM, Lacroux C, Douet JY, Benestad SL, Lantier F, et al. Evidence for zoonotic potential of ovine scrapie prions. Nat Commun. 2014;5:5821. doi: 10.1038/ncomms6821.
  2. Comoy EE, Mikol J, Luccantoni-Freire S, Correia E, Lescoutra-Etchegaray N, Durand V, et al. Transmission of scrapie prions to primate after an extended silent incubation period. Scientific reports. 2015;5:11573. doi: 10.1038/srep11573.
    • An additional sentence is now included that cites the two studies above.
    • Revision: Still, this low likelihood leaves room for the possibility of scrapie transmission to humans.
    • Lines 128-129

  • Line 517-518: Authors described that the withe-tailed deer wild-type prion protein allele is Q95/S96… I think that Q95/G96 is correct.
    • Corrected to Q95/G96
    • Line 524
  • RME in Table 2.3: Authors mentioned the distributions and PrPSc types in the cerebellar cortex of Rocky Mountain elk affected with CWD, and the data were from author’s lab. Therefore, RME in Table 2.3 should be filled out. Otherwise, tell me the reason why (-) not reported.
    • The details provided by the paper is limited to the cerebellar white matter, which is not in the table.
    • Data collected at the time of the original study indicates that the white matter was most affected, but does not further specify the molecular or granule cell layer of the cerebellar cortex.

  • iCJD in Table 2.4: At least, one report states that immunohistochemistry for PrP (3F4) showed a synaptic staining pattern in both the molecular and granular layers in cerebellar cortex of Dura mater graft-associated CJD patient (Shijo et al. 2017. Neuropathology). 
    • Descriptions are added to the Table 2.4.
    • Revision: One report on an individual with dura mater-derived iCJD states that there was Type 1 PrPSc accumulated in the cerebellar cortex. A synaptic staining pattern was found in both the molecular and granule cell layer of the cerebellar cortex.
    • Lines 563-566
  • Line 768: REF 198 is not appropriate. For example, Gill et al. 2013. BMJ may be an appropriate reference.
    • Reference has been replaced with Gill et al. 2013
    • Line 776
  • Line 1098: The title of REF 128 is something wrong. Please reconfirm the reference.
    • Reference has been updated to include full article title
